# *Escherichia coli* phylogeny drives co-amoxiclav resistance through variable expression of TEM-1 beta-lactamase

William Matlock [1,2,3,7] ✉, Gillian Rodger [2,3,7], Emma Pritchard [2,3], Matthew Colpus [3], Natalia Kapel [3], Lucinda Barrett [4], Marcus Morgan [4], Sarah Oakley [4], Katie L. Hopkins [5], Aysha Roohi [2,3], Drosos Karageorgopoulos [4], Matthew B. Avison [6], A. Sarah Walker [2,3], Samuel Lipworth [2,3,4,8] & Nicole Stoesser [2,3,4,8] ✉

Co-amoxiclav (amoxicillin and clavulanate) is a commonly used combination antibiotic, with resistance in *Escherichia coli* associated with increased mortality. The class A beta-lactamase $bla_{TEM-1}$ is often carried by resistant *E. coli* but exhibits high phenotypic heterogeneity, complicating genotype-phenotype predictions. We curated a dataset of $n = 377$ diverse *E. coli* isolates where the only acquired beta-lactamase was $bla_{TEM-1}$. We generated hybrid assemblies and co-amoxiclav minimum inhibitory concentrations (MICs), and $bla_{TEM-1}$ qPCR expression data for a subset ($n = 67/377$). We first tested whether intrinsic expression of $bla_{TEM-1}$ varied between *E. coli* lineages, for example, from regulatory system differences, which are challenging to genomically quantify. Using genotypic features, we built a hierarchical Bayesian model for $bla_{TEM-1}$ expression, controlling for phylogeny. Expression varied across the phylogeny, with some lineages (phylogroups B1 and C, ST12) expressing $bla_{TEM-1}$ more than others (phylogroups E and F, ST372). Next, we built a second model to predict isolate MIC from genotypic features, again controlling for phylogeny. Phylogeny alone shifted MIC past the clinical breakpoint in 19% (55/292) of isolates with greater-than-chance probability, mostly representing ST12, ST69 and ST127. A third causal model confirmed that phylogenetic influence on $bla_{TEM-1}$ expression drove variation in MIC. We speculate that intergenic variation underlies this effect.

The class A beta-lactamase $bla_{TEM-1}$ was first identified in 1965 in a clinical *Escherichia coli* isolate[1]. Originally, it was mobilised by two of the earliest named transposons, Tn*2* and Tn*3*, located on plasmids[2]. In the decades since, the genetic context of $bla_{TEM-1}$ has evolved[3], and

other mobile genetic elements such as IS*26*[4] and a diverse array of plasmids[5] contribute to its dissemination. At the time of writing, NCBI contains over 220,000 unique isolates carrying $bla_{TEM-1}$ distributed across 26 genera, including the common clinical pathogens

[1]Department of Biology, University of Oxford, Oxford, UK. [2]Nuffield Department of Medicine, University of Oxford, Oxford, UK. [3]The National Institute for Health Research Health Protection Research Unit in Healthcare Associated Infections and Antimicrobial Resistance at the University of Oxford, Oxford, UK. [4]Oxford University Hospitals NHS Foundation Trust, Oxford, UK. [5]UK Health Security Agency, Colindale, UK. [6]School of Cellular and Molecular Medicine, University of Bristol, Bristol, UK. [7]These authors contributed equally: William Matlock, Gillian Rodger. [8]These authors jointly supervised this work: Samuel Lipworth, Nicole Stoesser. ✉e-mail: william.matlock@biology.ox.ac.uk; nicole.stoesser@ndm.ox.ac.uk

*Escherichia coli*, *Klebsiella pneumoniae* and *Acinetobacter baumannii*[6]. The emergence and dissemination of beta-lactam resistance has been a major healthcare challenge[7], and $bla_{TEM-1}$ represents a key example.

In the UK, beta-lactam and beta-lactamase inhibitor combinations such as co-amoxiclav (amoxicillin and clavulanate), are commonly used as a first-line treatment for severe infections[8]. For Enterobacterales, the current EUCAST co-amoxiclav minimum inhibitory concentration (MIC) clinical breakpoint for resistance is 8/2 μg/mL[9] across all indications, with a recent study concluding that empiric co-amoxiclav treatment of *E. coli* bacteraemia with MICs >32/2 μg/mL was associated with significantly higher mortality[10]. However, the carriage of $bla_{TEM-1}$ is associated with high phenotypic heterogeneity, making genotype-phenotype predictions challenging.

Small-scale, experimental $bla_{TEM-1}$ systems have demonstrated that the interplay of location (plasmid or chromosome) and copies in the genome, through varying dosage, contributes to variable resistance[11,12]. In addition, other determinants such as mutations in the promoter of $bla_{TEM-1}$[13] and the chromosomally intrinsic *ampC* gene[14], and efflux pumps[15], are associated with *E. coli* beta-lactam resistance. Moreover, different regulatory systems[16,17], epistasis (interaction between genes)[18,19] and epigenetics (heritable phenotypic changes without alterations to the underlying DNA sequence)[20], might also influence co-amoxiclav resistance. For example, five different *E. coli* strains carrying the same pLL35 plasmid (which carries $bla_{CTX-M-15}$ and $bla_{TEM-112}$) varied in cefotaxime resistance[21]. Likewise, the introduction of a pOXA-48, a common conjugative plasmid in carbapenem-resistant clinical Enterobacterales, to six different *E. coli* strains resulted in variable co-amoxiclav resistance[22]. This indicates that strain background plays a role in resistance.

Successful genotype-to-phenotype prediction requires a comprehensive understanding of not only individual resistant determinants but also their combined effects. Moreover, this understanding must be translated to clinically relevant pathogens. Yet, to accurately model resistance in these systems, a large sample of linked genomic and phenotypic data is required, which until recently has been limited by sequencing technology and costs.

In this study, we curated and completely reconstructed the genomes of 377 clinical *E. coli* bacteraemia isolates to reflect a real-world but relatively simple genetic scenario where the only acquired beta-lactamase gene identified was $bla_{TEM-1}$, all identical at the amino acid level. We quantified the co-amoxiclav MICs for these isolates and generated $bla_{TEM-1}$ qPCR expression data for a subset. We then modelled $bla_{TEM-1}$ expression and co-amoxiclav MIC whilst controlling for confounding genetic mechanisms and chromosomal phylogeny, and characterised differences in intergenic content between lineages that may be contributing to differential phenotypic effects.

## Results

### A curated dataset of *E. coli* isolates with hybrid assemblies and co-amoxiclav MICs

We began with $n = 548$ candidate *E. coli* isolates, which following hybrid assembly, were curated into a final dataset of $n = 377/548$ (see "Methods" and Supplementary Information). In total, 77% (291/377) of assemblies were complete (all contigs were circularised), with the remaining 27% (86/377) having at least a circularised chromosome to confidently distinguish between chromosomal and plasmid-associated $bla_{TEM-1}$. Assemblies contained median = 3 (IQR = 2–5) plasmid contigs.

We identified $n = 451$ $bla_{TEM-1}$ genes on 431 contigs (13% [58/431] chromosomal versus 87% [373/431] plasmid). Isolates carried a median = 1 copy of $bla_{TEM-1}$ (range = 1–6). Carrying more than one copy of $bla_{TEM-1}$ on a single contig was rare: of all $bla_{TEM-1}$-positive contigs, 97% (400/412) versus 3% (12/412) had no duplications versus at least one. The $bla_{TEM-1}$ genes had synonymous single-nucleotide polymorphisms (SNPs) in positions 18, 138, 228, 396, 474, 705 and 717, totalling $n = 7$ single-nucleotide variant (SNV) profiles across the replicons, yet

diversity was dominated by $bla_{TEM-1b}$ at 73% (329/451; SNV profile TATTTCG; see "Methods")[23]. Where $bla_{TEM-1}$-positive contigs had at least two copies of $bla_{TEM-1}$, they were almost always the same SNV duplicated (11/12).

By examining the genomic arrangement of $bla_{TEM-1}$ (namely the replicons it was found on as well as any copies), we found most isolates carried a single non-chromosomal copy (73% [277/377]; Fig. 1a, b). More generally, whilst the plasmid contigs totalled only 3.6% of the total sequence length (bp) across the assemblies (71,126,646 bp/1,969,804,202 bp), they carried 85.8% of the $bla_{TEM-1}$ genes (387/451). Such $bla_{TEM-1}$-carrying plasmids were represented across the *E. coli* phylogeny (Fig. 1c). Overall, the dataset comprised 5.6% (21/377) phylogroup A, 8.0% (30/377) B1, 48.5% (183/377) B2, 4.8% (18/377) C, 27.3% (103/377) D, 0.5% (2/377) E, 4.2% (16/377) F and 1.1% (4/377) G. In total, we manually corrected $n = 5$ EzClermont phylogroup classifications using a chromosomal core gene phylogeny (see "Methods"): OXEC-108 (G to D), OXEC-317 (B2 to D), OXEC-333 (U to B1), OXEC-344 (U to B1) and OXEC-406 (U to B1). The EzClermont publication presented a 98.4% (123/125) true-positive rate on their validation set, which is in line with our 98.7% (372/377)[24].

Five known upstream promoters modulate the expression of $bla_{TEM-1}$: *P3*, *Pa/Pb*, *P4*, *P5* and *Pc/Pd* [13,25]. We linked 91% (409/451) $bla_{TEM-1}$ genes to a promoter immediately upstream, of which a majority, 64% (262/409), were identical to the *P3* reference. More generally, excluding $n = 2$ different *Pc/Pd*-like promoters which have large deletions, we identified SNPs in positions 32, 43, 65, 141, 162 and 175, totalling $n = 8$ SNV profiles (by Sutcliffe numbering[26]). Notably, 15% (39/262) of promoters had the *Pa/Pb*-associated C32T mutation, which produces two overlapping promoter sequences. Figure S1 visualises the joint distribution of isolate phylogroup and linked $bla_{TEM-1}$ promoter SNV.

Isolates were associated with a diverse range of co-amoxiclav MICs (μg/mL; ≤2/2 [4 (1.1%)], 4/2 [24 (6.4%)], 8/2 [144 (38.2%)], 16/2 [86 (22.8%)], 32/2 [44 (11.7%)] and >32/2 [75 (19.9%)]; Fig. 1d; see "Methods"). Figure S2 visualises the joint distribution of isolate phylogroup and MIC. Figure S3 visualises the distribution of $bla_{TEM-1}$ genome and cell copy number, $bla_{TEM-1}$ expression, and co-amoxiclav MIC against the chromosomal core gene phylogeny.

### $bla_{TEM-1}$ associated with conjugative plasmids

Whilst chromosomal copies of $bla_{TEM-1}$ can remain with a lineage over time, plasmidic copies might come and go. This could give the host cell access to a transient boost in resistance without impeding long-term fitness.

Confining the analysis to circularised plasmids (1036/1512), in silico replicon typing revealed the most common plasmid families to be ColRNAI-like, Col156-like, B/O/K/Z-like and Col(MG828)-like at 11% (117/1036), 7.4% (77/1036), 6.8% (70/1036) and 5.2% (54/1036), respectively. All other plasmid families had fewer than 50 representatives. Figure S4 visualises the relationship between strain background (sequence type and phylogroup) and replicon types (PlasmidFinder output; see "Methods") for the circularised plasmids. Using a 2-sample test for equality of proportions with continuity correction, the $bla_{TEM-1}$-positive plasmids (333/1036) were significantly more likely to be putatively conjugative (85.9% [286/333]) compared to the $bla_{TEM-1}$-negative plasmids (28.4% [200/703]; prop = 200/703; $\chi^2 = 297.02$, df = 1, $p$-value < 2.2e-16). Moreover, for the most common genomic arrangement of $bla_{TEM-1}$ (i.e., a single plasmid [277/377]), amongst circularised plasmids (252/277), 90% were putatively conjugative (227/252).

For a genome carrying $bla_{TEM-1}$ on the chromosome, the gene's copy number and total number of genes in the genome are equivalent. For a genome carrying $bla_{TEM-1}$ on a plasmid, this might not be the case. This is because plasmids can exist as multiple copies. The calculated copy number of all plasmidic contigs ($n = 1512$) was median = 3.13

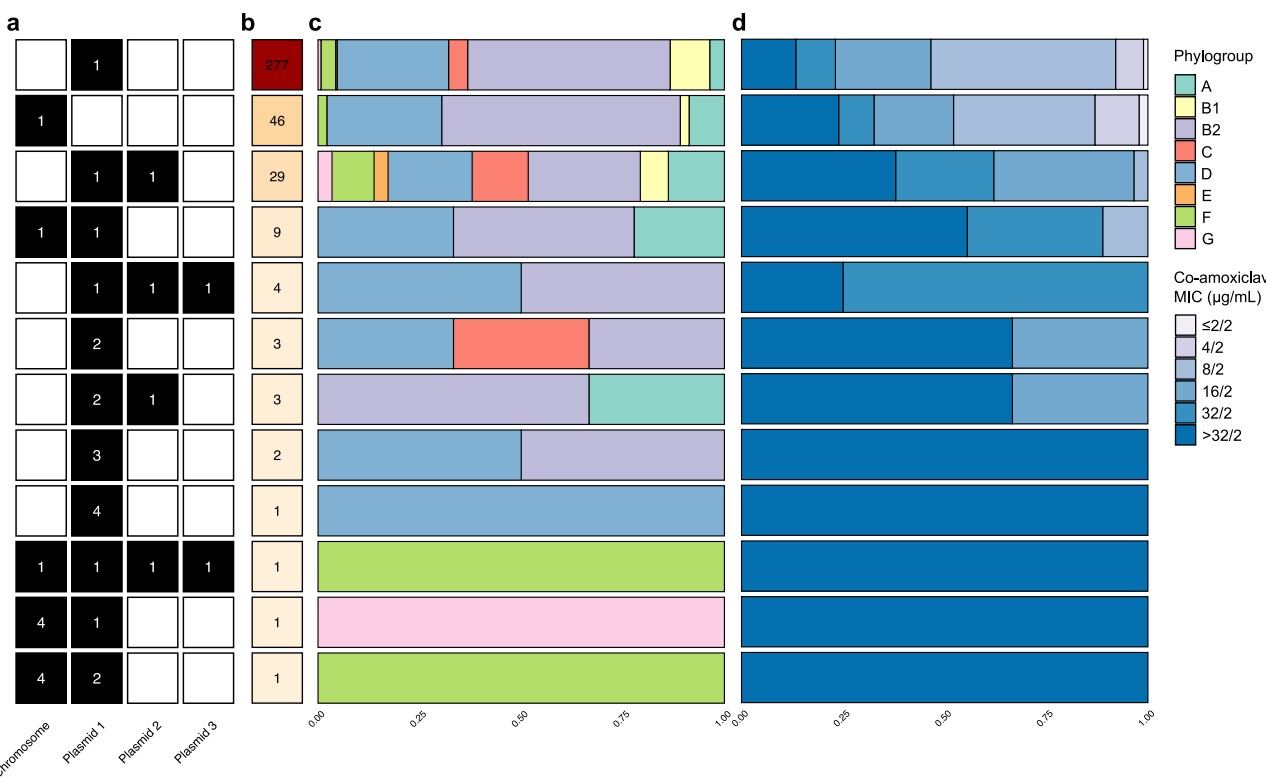

**Fig. 1 | A genotypically and phenotypically heterogenous population of $bla_{TEM-1}$-carrying *E. coli*. a** The genomic arrangement of $bla_{TEM-1}$ in the genomes ordered in descending (**b**) frequency. **c** Phylogroup and **d** co-amoxiclav MIC distribution for each genomic arrangement.

(range=0.04-57.00). Of these, $n = 19/1512$ contigs (with 6/19 circularised) had calculated copy numbers less than one (see "Methods"). This was potentially due to uneven short-read coverage. However, none carried $bla_{TEM-1}$ so were not used in the later modelling. Taking the circularised plasmids with copy number at least one (1030/1512), longer plasmids (>10kbp; 668/1030) were generally low copy number (median = 2.36), whilst shorter plasmids (≤10kbp; 362/1030) were generally high copy number (median = 11.01, see Fig. S5), consistent with previous studies[27].

### *E. coli* phylogeny shapes $bla_{TEM-1}$ expression

Within different *E. coli* lineages, $bla_{TEM-1}$ and its promoter are potentially subject to different regulatory systems and epigenetic interactions, which may in turn affect $bla_{TEM-1}$ expression. To test this, we first selected a random subsample of $n = 67/377$ isolates with a single copy of $bla_{TEM-1}$ in the genome, either on a chromosome (15/67) or a plasmid (52/67). Moreover, we only selected isolates with zero, one, or two mutations in the $bla_{TEM-1}$ promoter sequence: C32T, a well-studied but rare mutation which produces two overlapping promoters and is known to increase expression[13], and G175A, a less studied but common mutation in our dataset (according to Sutcliffe numbering based on the PBR322 plasmid[26]; see Table 1). Focussing on only two mutations enabled us to statistically explore the effect of their interaction. Isolates were distributed across the entire *E. coli* phylogeny (phylogroup A [6/67], B1 [4/67], B2 [40/67], C [4/67], D [8/67], E [1/67] and F ([4/67]). We then performed qPCR to evaluate for $bla_{TEM-1}$ expression (see "Methods"). Every isolate had at least two replicates (2 [48/67], 4 [1/67], or 9 [18/67]), giving a total of $n = 262$ $bla_{TEM-1}$ ΔCt observations (TEM-1 Ct – 16S Ct; see "Methods") for modelling.

To test for the effects of *E. coli* lineage, we built a maximum likelihood core gene phylogeny for all $n = 377$ chromosomes (see "Methods"). In total, we identified 17,836 gene clusters, of which 18.7% (3342/17,836) were core genes (those found in ≥98% of chromosomes). The phylogeny (midpoint-rooted and restricted to the $n = 67/377$

**Table 1 | Replicon distribution of $n = 67$ $bla_{TEM-1}$ promoter variants**

| Promoter SNV/ Replicon | Chromosome | Plasmid | Total |
|---|---|---|---|
| CG (wildtype) | 1 | 22 | **23** |
| G175A | 2 | 21 | **23** |
| C32T | 3 | 5 | **8** |
| C32T, G175A | 9 | 4 | **13** |
| Total | 15 | 52 | 67 |

Single-nucleotide variants (SNVs) are for the 32nd and 175th positions by Sutcliffe numbering[26].

isolates in the expression analysis) is given in Fig. 2a. Using $b = 1000$ ultrafast bootstraps, all phylogroup node supports were 100%, and more generally, 76.7% (287/374) of internal node supports were 100%, and 87.4% (327/374) were at least 95% (see "Methods"). Moreover, the Robinson-Foulds distance between the ML tree and consensus tree was 4, indicating nearly identical topology.

Briefly, the expression linear mixed model employed Markov Chain Monte Carlo (MCMC) to estimate parameters. The response variable $bla_{TEM-1}$ ΔCt (normalised and 95th percentile truncated) was related to the fixed effects (i) $bla_{TEM-1}$ cell copy number (normalised), (ii) presence of the C32T promoter mutation, (iii) presence of the G175A mutation, and (iv) their interaction. Random effects were incorporated to account for qPCR replicates and

phylogenetic relationships between isolates. See Supplementary Information for model specification, outputs and diagnostics.

In decreasing order of effect size, C32T, G175A, and a one unit increase in contig copy number all increased expression (decreased ΔCt; Table 2). There was no additional effect of G175A if C32T was also present (−1.69 < −1.71). After accounting for these covariates, we still identified a contribution from isolate phylogeny. The posterior

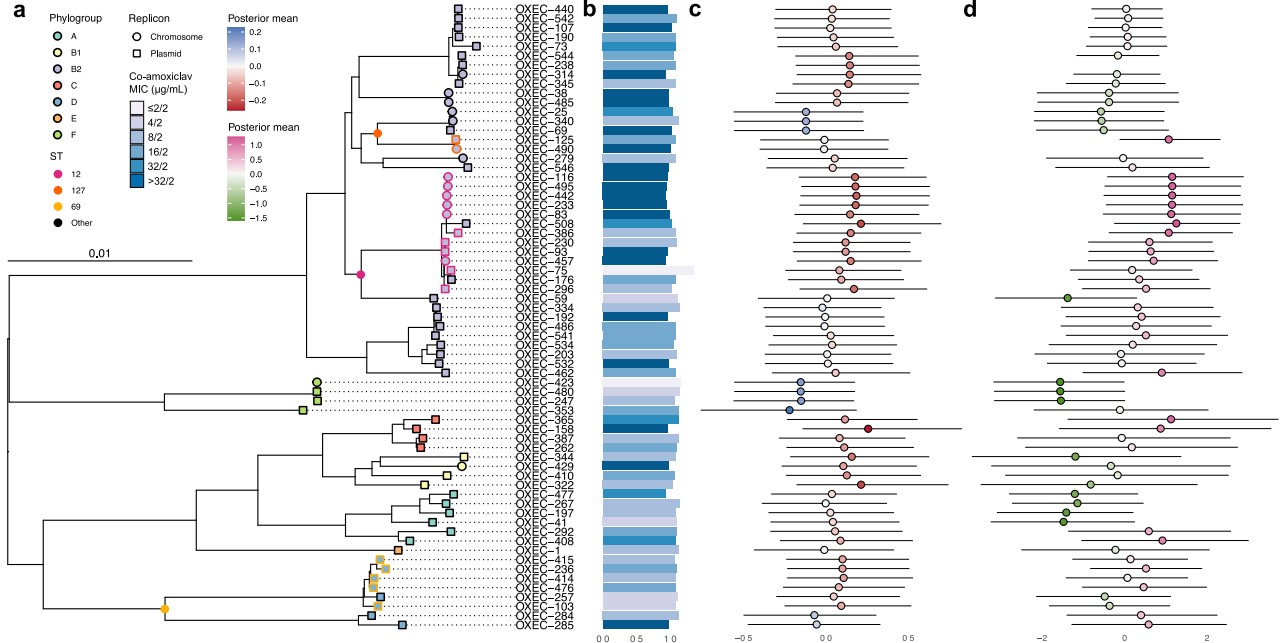

**Fig. 2 | Intrinsic expression of *bla*TEM-1 shapes co-amoxiclav MIC across the *E. coli* phylogeny. a** A midpoint-rooted core gene phylogeny of *E. coli* chromosomes from the 67 isolates with expression quantified. Tips are coloured by phylogroup. ST12, ST127 and ST69 are highlighted with tip outlines and circles at their crown nodes. Tip shape distinguishes location of *bla*TEM-1 on chromosomes (circles) and plasmids (squares). **b** Bar length records mean *bla*TEM-1 ΔCt for each tip (log10-scaled; lower values denote higher *bla*TEM-1 expression). Bar colour records isolate co-amoxiclav MIC for each tip. **c** Posterior means (coloured circles) and 95% HPD

intervals (horizontal lines) for phylogenetic effect on negative *bla*TEM-1 ΔCt for each tip (multiplied by −1 for ease of comparison). Red indicates above average expression and blue indicates below average expression. **d** Posterior means (coloured circles) and 95% HPD intervals (horizontal lines) for phylogenetic effect on co-amoxiclav MIC for each tip. Pink indicates above average MIC and green indicates below average MIC. Note that isolates OXEC-238 and OXEC-490 were excluded from the MIC model (see exclusion criteria).

**Table 2 | Parameter estimates for *bla*TEM-1 delta cycle threshold (ΔCt) genotype-phenotype model**

| Variable | Beta coefficient posterior mean | 95% HPD (*l*, *u*) | $p_{MCMC}$ |
|---|---|---|---|
| Intercept | 0.30 | 0.03, 0.59 | 0.0245 |
| C32T* | −1.71 | −2.08, −1.34 | <1e-05 |
| G175A* | −0.31 | −0.57, −0.04 | 0.0202 |
| C32T*:G175A* interaction | 0.33 | −0.14, 0.81 | 0.1779 |
| *bla*TEM-1 cell copy number | −0.12 | −0.24, 0.01 | 0.0679 |

All values were taken from chain 1. Lower (*l*) and upper (*u*) values are given for the 95% highest posterior density (HPD) intervals. $p_{MCMC}$ is the posterior probability that the coefficient was positive (defined in the Supplementary Information). The effect of both C32T* and G175A* is −1.71−0.31+0.33 = −1.69.
* Numbering by Sutcliffe[26].

for contribution of variance from phylogeny demonstrated a long right tail (mean = 0.07; 95% highest posterior density, HPD = [0.00, 0.21]; see "Methods"), suggestive of heterogeneity in phylogenetic signal, where deeper splits between major lineages may explain disproportionately large differences in expression. For qPCR replicates, the contribution of variance exhibited minimal skew (mean = 0.15; 95% HPD = [0.08, 0.23]). To investigate this further, we computed the posterior mean and 95% HPD credible interval for each tip in the phylogeny (Fig. 2c). Compared to the average across the *E. coli* phylogeny, some phylogroups (B1, C) and STs (12) were associated with increased *bla*TEM-1 expression, whilst some phylogroups (E, F) and STs (372) were associated with decreased *bla*TEM-1 expression.

## *ampC* gene variation is highly concordant with *E. coli* lineage

In many Gram-negative species, chromosomal *ampC* is regulated by the transcriptional activator AmpR and is inducible in the presence of beta-lactams. However, *E. coli* lacks *ampR* and therefore expresses *ampC* constitutively; overproduction depends on mutations in the promoter or attenuator regions. At the time of writing, the beta-lactamase database (BLDB) contains *n* = 4915 non-synonymous variants of the gene[28].

To quantify how well *ampC* variants agree with phylogroup and ST, we calculated the homogeneity (*h*) and completeness (*c*; both range from 0 to 1; see "Methods"). Briefly, *h* = 1 means that a phylogroup or ST contains a single *ampC* variant. Conversely, *c* = 1 means that all instances of an *ampC* variant fall within the same phylogroup or ST. For phylogroups, we found *h* = 0.489 and *c* = 0.964, and for STs (excluding 38/377, which were unassigned), *h* = 0.938 and *c* = 0.877. Overall, this suggests that phylogroups tend to contain distinct *ampC* variants, which are generally ST-specific, and overall, that *E. coli* phylogeny is a suitable proxy for *ampC* variation.

Whilst many *E. coli* *ampC* variants present a narrow spectrum of hydrolytic activity, some can potentially hydrolyse third-generation cephalosporins following mutations in the promoter sequence. To explore promoter variation, we aligned all *n* = 377 *ampC* promoter sequences. Mutations outside positions −42 to +37 (according to Jaurin numbering[14]) were disregarded based on existing characterisations[29,30]. In total, *n* = 12 *ampC* promoter SNVs were identified, with variation dominated by the *E. coli* K12 wildtype at 47% (177/377). Table 3 documents all *n* = 11 mutations identified. A given *ampC* variant associated almost uniquely with an *ampC* promoter variant, yet *ampC* promoter variants were associated with multiple *ampC* variants (*h* = 0.483 and *c* = 0.941).

**Table 3 | Variation in _n_ = 377 _ampC_ promoters**

| Region | Position | _E. coli_ K12 (_n_) | Mutation (_n_) |
|---|---|---|---|
| Spacer | −28 | G (243) | A (134) |
| | −18 | G (329) | A (48) |
| Between −10 box and attenuator | −1 | C (329) | T (48) |
| | +11 | T (371) | – (6) |
| Attenuator | +17 | C (316) | T (61) |
| | +22 | C (367) | T (10) |
| | +26 | T (367) | G (10) |
| | +27 | A (367) | T (10) |
| | +30 | G (376) | A (1) |
| | +32 | G (364) | A (13) |
| | +37 | G (376) | T (1) |

Positions are according to Jaurin numbering[14].

### _E. coli_ phylogeny drives co-amoxiclav resistance through expression

We next investigated whether the _E. coli_ lineages with intrinsically higher _bla_~TEM-1~ expression also had intrinsically higher co-amoxiclav MICs. This would be consistent with lineage differences in regulatory regions and epigenetic interactions driving increased resistance.

We employed an MCMC to estimate parameters in an ordinal mixed model. The response variable isolate co-amoxiclav MIC (μg/mL; levels ≤2/2, 4/22, 8/22, 16/2, 32/2, >32/2) was predicted by the fixed effects (i) _bla_~TEM-1~ cell copy number (normalised and 95th percentile truncated), (ii) _bla_~TEM-1~ genome copy number (>1 vs. 1), (iii) non-wildtype _bla_~TEM-1~ promoter SNVs, and (iv) non-wildtype _ampC_ promoter SNVs. For the model, we only used isolates for which every _bla_~TEM-1~ gene was linked to a promoter, and all the promoters were the same variant. We then filtered out isolates with _bla_~TEM-1~ promoter and _ampC_ promoter variants that appeared less than 10 times. This left _n_ = 292/377 isolates. Full model specification, convergence diagnostics, and outputs are given in Supplementary Information.

In decreasing order of effect size, the presence of C32T and G175A in the _bla_~TEM-1~ promoter, the presence of just C32T, _bla_~TEM-1~ cell copy number and _bla_~TEM-1~ genome copy number all increased co-amoxiclav MIC (Table 4); the remaining effects were compatible with chance. As with the expression model, the posterior distribution for contribution of variance from phylogeny demonstrated a long right tail (mean = 2.80; 95% HPD = [0.72, 5.16]). The phylogeny for the _n_ = 292 isolates with co-amoxiclav MIC tip effects is given in Fig. S6.

We found that phylogenetic tip effects alone were sufficient to shift category membership across the EUCAST resistance breakpoint (> 8/2 μg/mL). Fixing all other covariates at the midpoint of their latent categories, 19% (55/292) of isolates had a greater than 50% posterior probability of being shifted from the susceptible to the resistant category due to their phylogenetic effect alone. Of these, most were represented by phylogroup B2 and D at 64% (35/55) and 20% (11/55), respectively. The most represented sequence types were ST12, ST127 and ST69 at 27% (15/55), 20% (11/55) and 18% (10/55), respectively. The isolates in these sequence types also used in the expression model are highlighted in Fig. 2d.

Lastly, we developed a combined model to test whether the phylogenetic influence on _bla_~TEM-1~ expression causally varies co-amoxiclav MIC (see Supplementary Information). Here, we only used the predictors identified as significant from previous expression and MIC models (_bla_~TEM-1~ cell and genome copy numbers, and _bla_~TEM-1~ promoter SNV). Briefly, the model estimates a parameter that scales the phylogenetic and non-phylogenetic random effects from expression to MIC. Under causality, the scaling parameter should be constant across all random effect terms. We found the scaling parameter had a

**Table 4 | Parameter estimates for co-amoxiclav minimum inhibitory concentration (MIC) genotype-phenotype model**

| Variable | | Beta coefficient posterior mean | 95% HPD (_l_, _u_) | _p_~MCMC~ |
|---|---|---|---|---|
| Intercept | | 3.92 | 2.92, 4.85 | <1e-05 |
| _bla_~TEM-1~ cell copy number | | 2.07 | 1.38, 2.78 | <1e-05 |
| _bla_~TEM-1~ genome copy number (>1 vs. 1) | | 0.97 | 0.02, 1.90 | 0.0414 |
| _bla_~TEM-1~ promoter SNV vs. wildtype | G175A[a] | 0.17 | -0.35, 0.69 | 0.5313 |
| | C32T[a] | 6.06 | 4.15, 8.08 | <1e-05 |
| | C32T[a] and G175A[a] | 5.87 | 3.89, 7.87 | <1e-05 |
| _ampC_ promoter SNV vs. wildtype | G-28A[b] and C17T[b] | 0.44 | −0.63, 1.48 | 0.4122 |
| | G-28A[b] | 0.87 | −0.89, 2.58 | 0.3039 |
| | G-18A[b] and C-1T[b] | 0.83 | −1.37, 3.11 | 0.4402 |

All values were taken from chain 1. For single-nucleotide variant (SNV) vs. wildtype, _E. coli_ K12 was used as the wildtype baseline. Lower (_l_) and upper (_u_) values are given for the 95% highest posterior density (HPD) intervals. _p_~MCMC~ is the posterior probability that the coefficient was positive (defined in the Supplementary Information).
[a] Numbering by Sutcliffe[26].
[b] Numbering by Jaurin[14].

posterior mean = −1.13 (95% HPD = [−0.72, −1.49]; _p_~MCMC~ = 0.002; values taken from chain 1). This supports a direct and substantial influence of expression on MIC, mediated by phylogenetic relationships.

### Intergenic architecture and dynamics vary between _E. coli_ phylogroups

The results from our modelling suggest that lineage-specific regulatory systems play a role in the expression of _bla_~TEM-1~, which in turn modulates resistance. The intergenic regions (IGR) in bacterial chromosomes contain their regulatory systems, with one study finding that 86% of IGRs in a strain of _E. coli_ were transcriptionally active[31]. Like genes, IGRs are subject to genetic flow within bacterial populations. Yet, how closely these dynamics mirror one another remains unclear. Understanding whether IGRs evolve more rapidly than coding sequences can shed light on how bacterial populations adapt and diversify not just at the level of the genes they carry, but in how they control the expression of those genes, ultimately affecting their phenotypic traits.

We first began with a traditional pangenome analysis for all _n_ = 377 chromosomes (see "Methods"). Here, we identified _n_ = 15,781 gene clusters, with 5.0% (782/15,781) present in all isolates, and singletons (one member) and doubletons (two members) comprising 22.7% (3576/15,781) and 9.0% (1413/15,781), respectively. We then performed a similar analysis, but for IGR clusters (see "Methods"). In total, we identified _n_ = 33,345 IGR clusters, of which 1.0% (322/33,345) were present in all isolates, consistent with a small core IGR system. Most IGR clusters represented singletons at 44.7% (14,903/33,345) or doubletons at 11.9% (3977/33,345). We then visualised the top 10,000 IGR clusters against our _E. coli_ phylogeny in Fig. 3a. This indicated phylogroup- and sequence type-specific patterns in intergenic sequence.

We next wanted to understand whether IGR diversity outpaced coding-sequence diversity. To do this, we built cluster accumulation curves, randomising genome-addition order 100 times per phylogroup, then plotting the cumulative number of clusters discovered versus the number of genomes sampled. We performed this for all phylogroups with at least 20 members (A, B1, B2 and D) in two separate runs: for all cluster sizes, and clusters with at least two members to

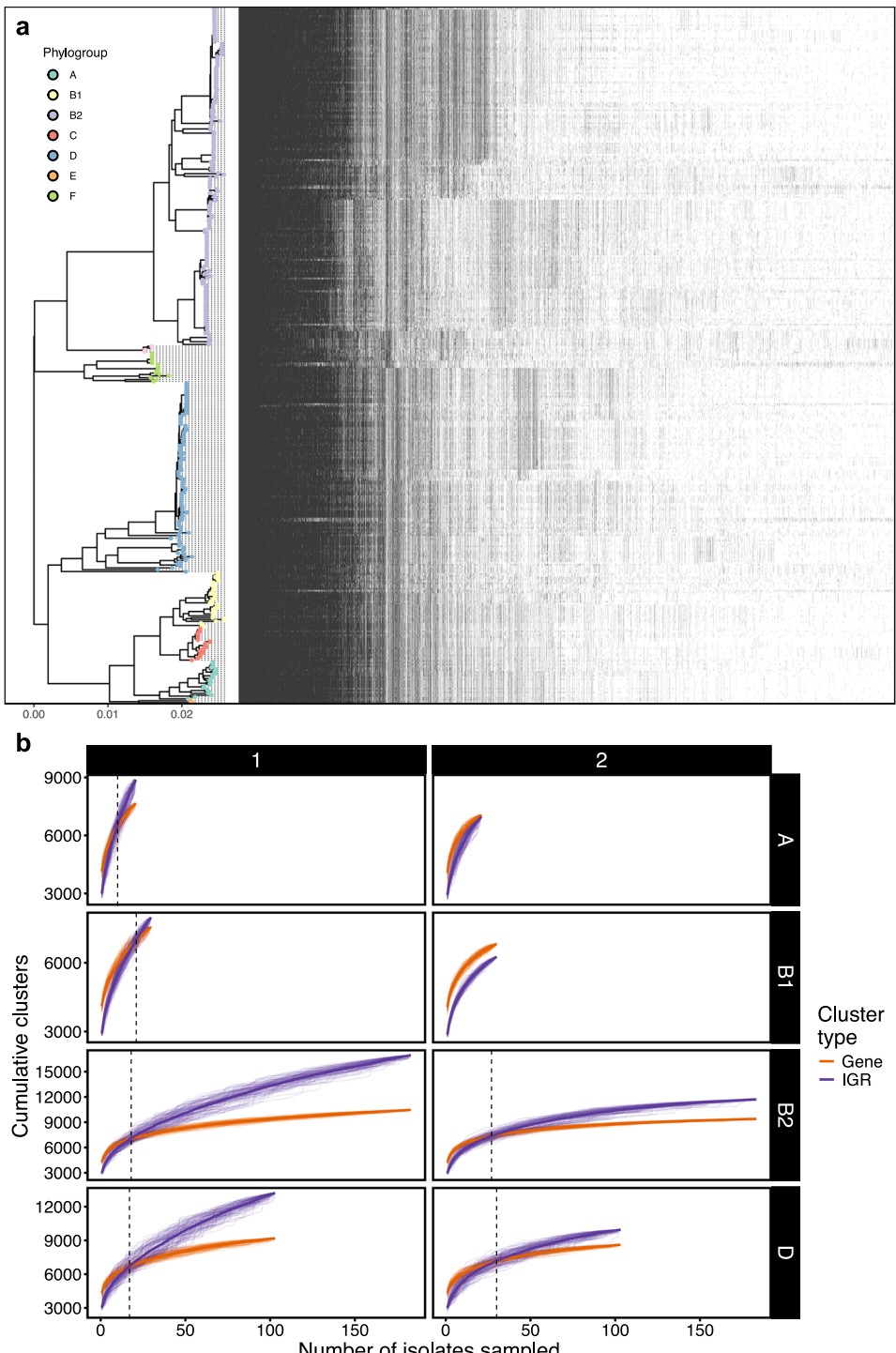

**Fig. 3 | Intergenic region content and dynamics vary across the *E. coli* phylogeny. a** The top 10,000 most common IGR clusters ordered by prevalence (left to right), arranged by midpoint-rooted core gene phylogeny. Tips are coloured by phylogroup. **b** Gene and IGR cluster accumulation curves for phylogroups A, B1, B2 and D. Dashed vertical lines indicate crossover points of averaged accumulation curves.

account for possible assembly and bioinformatic errors. The averaged curves are presented in Fig. 3b. When considering clusters of all sizes, the IGR curves began below the gene curves, reflecting rapid discovery of moderately common accessory genes, but all overtook them after a period of sampling. This crossover marks the sampling depth at which IGR novelty becomes the dominant source of new information.

Fitting Heaps' law to the mean curves provided a concise measure of openness (see Fig. 3b and Table 5). Briefly, $\alpha$ describes the rate at which diversity accumulates as more samples are taken, where a higher $\alpha$ indicates faster accumulation. We found that $\alpha$ was consistently higher for IGR curves than for gene curves, and phylogroup A exhibited the highest $\alpha$, indicative of the most open IGR system. To ensure these patterns were not driven by unequal sampling, we regressed $\alpha$ against both phylogroup sample size and cluster type (IGR vs gene). In this model (adjusted $R^2 = 0.73$), sample size had no significant effect on $\alpha$ ($\beta = -1.2\text{e-}04$, $p$-value $= 0.48$), whereas IGRs showed a highly

**Table 5 | Heaps' law model alpha (α) estimates for gene and intergenic region (IGR) cluster analysis**

| Phylogroup | Minimum cluster size | Cluster type | α |
|---|---|---|---|
| A | 1 | IGR | 0.39 |
|  |  | Gene | 0.21 |
|  | 2 | IGR | 0.30 |
|  |  | Gene | 0.18 |
| B1 | 1 | IGR | 0.32 |
|  |  | Gene | 0.20 |
|  | 2 | IGR | 0.23 |
|  |  | Gene | 0.16 |
| B2 | 1 | IGR | 0.36 |
|  |  | Gene | 0.17 |
|  | 2 | IGR | 0.26 |
|  |  | Gene | 0.14 |
| D | 1 | IGR | 0.35 |
|  |  | Gene | 0.17 |
|  | 2 | IGR | 0.27 |
|  |  | Gene | 0.15 |

α represents the decay parameter from fitting Heaps' law model. Contains estimates for *E. coli* isolate phylogroups A ($n = 21$), B1 ($n = 30$), B2 ($n = 183$) and D ($n = 103$).

significant positive shift ($\beta = 0.137$, *p*-value = 2.2e-05). Thus, the faster accumulation of IGR diversity truly reflects biological differences in chromosomal IGR dynamics, not sampling artifacts.

## Discussion

In our dataset of clinical *E. coli*, $bla_{TEM-1}$ was overwhelmingly carried by conjugative plasmids. This means it can spread between bacterial hosts and different genetic backgrounds. We demonstrated that different bacterial hosts intrinsically vary in their ability to express $bla_{TEM-1}$ when accounting for variation in promoters (C32T and G175A mutations) and contig copy number. Moreover, our findings suggest that some clinically successful lineages (e.g., ST12) are better at expressing $bla_{TEM-1}$ than less clinically successful lineages (e.g., phylogroup F). With a second model, we also found that different *E. coli* lineages vary intrinsically in co-amoxiclav MIC (accounting for $bla_{TEM-1}$ genome and cell copies, and $bla_{TEM-1}$ and *ampC* promoter variants). Again, we observed that some clinically successful lineages (e.g., ST12) had higher resistance than less clinically successful lineages (e.g., phylogroup F). We also quantified that the clinically successful sequence types ST12, ST69 and ST127 had the highest probability of flipping an isolate from the sensitive to resistant category. A third model demonstrated that these two traits were causally linked: *E. coli* phylogeny drives co-amoxiclav resistance through variable expression of $bla_{TEM-1}$, and underscores the necessity of fully resolving bacterial genomes to incorporate accurate genetic, genomic, and phylogenetic information in resistance prediction models. Future work could include evaluations of single amino acid substitutions in TEM-1 (that hydrolyse third-generation cephalosporins and carbapenemases[28]), which are typically carried in more complex genetic backgrounds.

This study has limitations. Firstly, it is possible that, due to fragmented plasmid assemblies, some isolates identified as having multiple copies of $bla_{TEM-1}$ on multiple plasmids instead had multiple copies on the same plasmid. Nonetheless, our expression analysis only considered isolates with a single copy of $bla_{TEM-1}$ in the genome, mitigating this concern. Secondly, we only examined expression for a subsample of our isolates due to resource limitations. Thirdly, whilst there is not an agreed upon standard reference gene for quantifying beta-lactamase expression[32–34], previous work has shown 16S to be

stable[33]. Crucially, our delta Ct values were consistent within isolates. Fourthly, we only observed two potentially relevant porin mutations (a premature stop codon in OmpC on OXEC-40's chromosome and in OmpF on OXEC-423's chromosome), limiting our ability to investigate their effects on phenotype. We also identified no relevant efflux pumps beyond AcrEF-TolC (found in every isolate), and it was out of the scope of the study to explore their functionality. Fifthly, we found that tip effects did not strictly align with phylogenetic structure. We used a single fixed phylogeny and a single global phylogenetic-variance parameter, which does not capture uncertainty in tree topology or clade-specific evolutionary rates. Consequently, tips within densely sampled clades can be over-shrunk toward their mean, while tips in sparse clades can be under-shrunk. Future work should incorporate phylogenetic uncertainty (e.g., sampling from a tree posterior) and explore multi-variance or partitioned phylogenetic models to assess the impact of tree balance on tip-effect estimation. Sixthly, modelling phylogenetic effect terms for all the plasmid replicons in the dataset would be computationally prohibitive and statistically unstable. Whilst future studies could focus on curating datasets with less plasmid diversity, this would not be reflective of true clinical bacterial populations. Penultimately, automated susceptibility testing methods, like the BD Phoenix™ used here, may not agree completely with reference methods; yet previous work has shown strong agreement with the EUCAST agar dilution method[35]. Lastly, plasmid copy number is not static. Moreover, in the presence of antibiotics, it has been demonstrated that resistance gene-carrying plasmids can increase their copy number to increase the chance of survival[36]. Our point estimates of plasmid copy number were derived from genome assemblies sequenced in the absence of antibiotics, which likely represent a lower bound. Nonetheless, we found strong signal to suggest the import of plasmid copy number on resistance, even if under our sensitivity testing, plasmid copy number potentially increased within isolates.

We posit that lineage-specific regulatory systems in *E. coli*, shaped by horizontal gene flow, may house the key modulators of $bla_{TEM-1}$ expression. Although comprehensive dissection of these elements (for example, through ChIP-seq) extends beyond the scope of this study, our findings suggest that future efforts directed at mapping trans-acting factors, small RNAs, differential DNA methylation and nucleoid-associated protein binding across phylogroups will be essential. Additionally, examining other resistance genes and their expression patterns in a similar phylogenetic framework could provide a broader understanding and prediction of resistance mechanisms across different bacterial species and antibiotics. Our study demonstrates the some clinically successful lineages are better at expressing $bla_{TEM-1}$, and have a higher probability of flipping from sensitive to resistant. We speculate that, since $bla_{TEM-1}$ is both widespread in clinical bacterial populations and its spread is plasmid mediated, being able to better control the expression of $bla_{TEM-1}$ when acquired is a selective advantage for clinical isolates.

## Methods
### Ethics oversight
The use of genotypic and phenotypic data from these isolates is covered by ethical permissions (London−Queen Square Research Ethics Committee, REC ref. 17/LO/1420). Isolates were a subset of those evaluated in a previous study[37], for which data linkage with patient data/antibiotic susceptibility test data was enabled through the Infections in Oxfordshire Research Database (IORD). This database has generic Research Ethics Committee, Health Research Authority and Confidentiality Advisory Group approvals (19/SC/0403, 19/CAG/0144) which facilitate the pseudo-anonymised linkage of routinely collected NHS electronic healthcare record data from the Oxford University Hospitals NHS Foundation Trust Clinical Systems Data Warehouse and research data (e.g., sequencing data) from the Modernising Microbiology and Big Infection Diagnostics Theme of the Oxford NIHR

Biomedical Research Centre, Oxford. IORD links records by a specific, random, number ensuring that no patient-identifiable information is shared with researchers using this resource.

## Isolate selection

We considered $n = 548$ candidate *E. coli* bacteraemia isolates cultured from patients presenting to Oxford University Hospitals NHS Foundation Trust between 2013 and 2018, and selected from a larger study of systematically sequenced isolates based on screening their short-read only assemblies with NCBIAMRFinder (v. 3.11.2) for $bla_{TEM-1}$, and the absence of other beta-lactamases[38].

## DNA extraction and sequencing

Sub-cultures of isolate stocks, stored at −80 °C in 10 % glycerol nutrient broth, were grown on Columbia blood agar (CBA) overnight at 37 °C. DNA was extracted using the EasyMag system (bioMerieux) and quantified using the Broad Range DNA Qubit kit (Thermo Fisher Scientific, UK). DNA extracts were multiplexed as 24 samples per sequencing run using the Oxford Nanopore Technologies (ONT) Rapid Barcoding kit (SQK-RBK110.96) according to the manufacturer's protocol. Sequencing was performed on a GridION using version FLO-MIN106 R9.4.1 flow cells with MinKNOW software (v. 21.11.7) and basecalled using Guppy (v. 3.84). Short-read sequencing was performed on the Illumina HiSeq 4000, pooling 192 isolates per lane, generating 150 bp paired end-reads[37].

## Dataset curation and genome assembly

Full details are given in Supplementary Information. Briefly, short- and long-read quality control used fastp (v. 0.23.4) and filtlong (v. 0.2.1), respectively[39,40]. We also used Rasusa (v. 0.7.1) on $n = 3/548$ long-read sets due to memory constraints[41]. Genome assembly used Flye (v. 2.9.2-b1786) with bwa (v. 0.7.17-r1188) and Polypolish (v. 0.5.0), and Unicycler (v. 0.5.0), which used SPAdes (v. 3.15.5), miniasm (v. 0.3-r179) and Racon (v. 1.5.0)[42–48]. Plasmid contig validation used Mash screen (v. 2.3) with PLSDB (v. 2023_06_23_v2)[49,50]. All assemblies were annotated with NCBIAMRFinder (v. 3.11.26 and database v. 2023-11-15.1)[38]. Alongside, we validated the presence of $bla_{TEM-1}$ using tblastn (v. 2.15.0+) with the NCBI Reference Gene Catalog TEM-1 RefSeq protein WP_000027057.1 and 100% amino acid identity[51]. Following genome assembly, we removed $n = 171/548$ isolates, either because (i) the chromosome did not circularise (116/171), (ii) it carried a non-$bla_{TEM-1}$ $bla_{TEM}$ variant and/or an additional acquired beta-lactamase (54/171), or (iii) the chromosome was too short consistent with misassembly (~ 3.5Mbp; 1/171). This left a final dataset of $n = 377$ isolates.

## Antibiotic susceptibility testing

Antibiotic susceptibility testing was performed using the BD Phoenix™ system in accordance with the manufacturers' instructions, generating MICs for co-amoxiclav.

## Generation of cDNA template

RNA extraction and DNase treatment were performed on replicates of each isolate ($n = 3$ biological/$n = 3$ technical) as described previously[52]. RNA was quantified post DNase treatment using Broad Range RNA Qubit kit (Thermo Fisher Scientific, UK), normalised to 1 μg and reverse transcribed to cDNA using SuperScript IV VILO (Thermo Fisher Scientific, UK) under the following conditions: 25 °C for 10 min, 42 °C for 60 min and 85 °C for 5 min.

## qPCR quantification of $bla_{TEM-1}$ expression

$bla_{TEM-1}$ expression was quantified in a selection of isolates: initially $n = 35$ isolates in triplicate, referred to as batch 1; then a further $n = 48$ isolates in duplicate, referred to as batch 2. Batch 1 were randomly selected by MIC ($n = 2$ MIC ≤2/2, $n = 5$ MIC 4/2, $n = 9$ MIC 8/2, $n = 10$ MIC 16/2, $n = 4$ MIC 32/2, $n = 5$ MIC > 32/2). Batch 2 was enhanced for specific $bla_{TEM-1}$ promoter mutations, selecting all isolates with a single $bla_{TEM-1}$ gene with C32T (with or without a G146A mutation) that had not already been tested, and then randomly selecting from other wildtype and G146A, single $bla_{TEM-1}$ gene isolates. For all qPCR reactions, *E. coli* cDNA was normalised to 1 ng and amplified in a duplex qPCR reaction targeting $bla_{TEM-1}$ and 16S. qPCR standard curves were prepared for both $bla_{TEM-1}$ (Genbank Accession: DQ221255.1) and 16S (Genbank Accession: LC747145.1) sequences cloned into pMX vectors (Thermo Fisher Scientific, UK). Tenfold dilutions of linearised plasmids ($1–1 \times 10^7$ copies/reaction) were used as a standard curve for each experiment. Both curves were linear in the range tested (16S: $R^2 > 0.991$; TEM-1: $R^2 > 0.91$). The slopes of the standard curves for 16S and $bla_{TEM-1}$ were −3.607 and −3.522, respectively. qPCR was performed using a custom 20 μl TaqMan gene expression assay consisting of TaqMan™ Multiplex Master Mix, TaqMan unlabelled primers and a TaqMan probe with dye label (FAM for TEM-1 and VIC for 16S) carried out on the QuantStudio5™ real-time PCR system (Thermo Fisher Scientific, UK). Cycling conditions were 95 °C for 20 s, followed by 40 cycles of 95 °C for 3 s and 60 °C for 30 s, with Mustang purple as the passive reference. For batch 1, triplicate samples were analysed and standardized against 16S rRNA gene expression. Triplicate reactions for each isolate demonstrated good reproducibility for batch 1 (Fig. S7). Of note, for isolate OXEC-75, TEM-1 expression was very low-level, and 5 reactions (1 technical replicate for biological replicate 1, 1 technical replicate for biological replicate 2, and all 3 technical replicates for biological replicate 3) failed to amplify any product. Due to resource constraints, we reduced replicates for batch 2 ($n = 1$ biological/$n = 2$ technical; Fig. S8). To reduce model complexity, we omitted some batch 1 isolates ($n = 16/35$) which carried more than one copy of $bla_{TEM-1}$ in the genome, leaving a total of $n = 67$ isolates. ΔCt values were calculated by subtracting mean 16S Ct from mean TEM-1 Ct.

## Assembly annotations

We annotated the chromosomes using Prokka (v. 1.14.6) with default parameters except `--centre X --compliant` (see `annotate.sh`)[53]. Abricate (v. 1.0.1) was used with default parameters and the PlasmidFinder database (v. 2023-Nov-4) to annotate for plasmid replicons[54,55]. Plasmid mobilities were predicted using MOB-suite's MOB-typer (v. 3.1.4) with default parameters[55]. Briefly, a plasmid was labelled as putatively conjugative if it had both a relaxase and mating pair formation (MPF) complex, mobilisable if it had either a relaxase or an origin of transfer (oriT) but no MPF, and non-mobilisable if it had no relaxase and oriT. Lastly, we assigned sequence types (STs) and phylogroups to our *E. coli* chromosomes using mlst (v. 2.23.0) with default parameters and EzClermont (v. 0.7.0) with default parameters, respectively[24,56]. We used blastn (v. 2.15.0+) with a custom database of known $bla_{TEM-1}$ promoters[13,25,57]. Due to the high similarity between the *P3*, *Pa/Pb*, *P4* and *P5* reference sequences, we chose the top hit in each position.

## SNV analysis

We first determined the sets of sequences we wanted to align: (i) $bla_{TEM-1}$ ($n = 451$; some genomes carried multiple copies), (ii) $bla_{TEM-1}$ promoters ($n = 409$), (iii) *ampC* ($n = 377$) and (iv) *ampC* promoters ($n = 377$). For $bla_{TEM-1}$ and *ampC*, we extracted the relevant sequences using the coordinate and strand information from the NCBIAMRFinder output (see `extractGene.py`). For the $bla_{TEM-1}$ promoters, we used coordinate and strand information from the earlier blastn results. For the *ampC* promoters, we took the sequence 200 bp upstream of the *ampC* gene then manually excised the −42 to +37 region in AliView[58]. Sets of sequences were aligned using MAFFT (v. 7.520) with default parameters except `--auto`[59]. Variable sites were examined using snp-sites (v. 2.5.1) with default parameters and in `-v` mode[60].

## Contig copy number

We used BWA (v. 0.7.17-r1188) to map the quality-controlled short-reads to each contig, then SAMtools (v. 1.18) for subsequent processing (see `copyNumber.sh`)[43,61]. For each contig, we calculated the mean depth over its length, then within each assembly, normalised by the mean depth of the chromosome.

## Chromosomal core gene phylogeny

Building the chromosomal phylogeny involved four main steps: annotating the chromosomes, identifying the core genes, aligning them and building a phylogeny. Initially, all the chromosomes carried a copy of *ampC*, meaning it was a core gene and would be included in the phylogeny. Since we wanted to manually verify EzClermont phylogroup classifications with the phylogeny and then compare phylogroups to the distribution of *ampC* gene variants, we excised the *ampC* sequence from all the chromosomes beforehand to avoid confounding our analysis (see `removeGene.py`). To identify the core genes (those with ≥98% frequency in the sample), we used Panaroo (v. 1.4.2) with default parameters except `--clean-mode sensitive --aligner mafft -a core --core_threshold 0.98`[62]. Panaroo also aligned our core genes using MAFFT (v. 7.520; see `runPanaroo.sh`)[59]. Lastly, we built the core gene maximum-likelihood phylogeny using IQ-Tree (v. 2.3.0) with default parameters except `-m GTR + F + I + R4 -keep-ident -B 1000 -mem 10 G` using `-s core_gene_alignment_filtered.aln` from Panaroo (see `runIQTREE.sh`)[63]. The substitution model used was general time reversible (GTR) using empirical base frequencies from the alignment (F), allowing for invariant sites (I) and variable rates of substitution (R4).

## Intergenic region analysis

For the gene cluster analysis, we used Panaroo (v. 1.5.1) with default parameters except `--clean-mode sensitive --merge_paralogs`[62]. Running Panaroo with paralog merging is necessary for Piggy, and reduced the overall pangenome size from the previous run. For the intergenic region cluster analysis, we used Piggy (v. 1.5) with default parameters[64].

## Statistical analysis and visualisation

All statistical analysis was performed in R (v. 4.4.0) using RStudio (v. 2024.04.2+764)[65,66]. We implemented MCMC generalised linear mixed models using the MCMCglmm library in R[67]. Model specifications, convergence diagnostics, parameter estimations and outputs are reported in Supplementary Information; see `modelExpression.R`, `modelMIC.R` and `modelCombined.R`, to reproduce the *bla*TEM-1 expression, co-amoxiclav MIC and causal models, respectively. Homogeneity and completeness are defined in Rosenberg, A. and Hirschberg, J (2007) and were also implemented in R[68]. A 95% highest posterior density (HPD) credible interval finds the closest points ($a$ and $b$) for which $F(b) - F(a) = 0.95$, where $F$ is the empirical density of the posterior. Figures were plotted with the ggplot2 library[69]. See `buildResults.R` and `igr_analysis.R` to reproduce all statistics and figures in the manuscript.

## Reporting summary

Further information on research design is available in the Nature Portfolio Reporting Summary linked to this article.

## Data availability

Metadata for all $n = 377$ genomes included in the final analysis is given in Supplementary Data File 1. Metadata for all $n = 451$ *bla*TEM-1 annotations identified in these genomes is given in Supplementary Data File 2. qPCR expression data for all replicates is given in Supplementary Data File 3. NCBI accessions for short- and long-read sets and assemblies are given in Supplementary Data File 4. All Supplementary Data Files are also stably archived at https://doi.org/10.5281/zenodo.16731807.

## Code availability

All scripts referenced in the Methods can be found in the GitHub repository https://github.com/wtmatlock/tem, which is stably archived at https://doi.org/10.5281/zenodo.16731807.

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

## Acknowledgements

This work was funded by the National Institute for Health Research (NIHR) Health Protection Research Unit in Healthcare Associated Infections and Antimicrobial Resistance (NIHR200915), a partnership between the UK Health Security Agency (UKHSA) and the University of Oxford. It was also supported by the NIHR Oxford Biomedical Research Centre (BRC). The computational aspects of this research were funded from the NIHR Oxford BRC with additional support from the Wellcome Trust Core Award Grant Number 203141/Z/16/Z. The views expressed are those of the author(s) and not necessarily those of the NIHR, UKHSA or the Department of Health and Social Care. The authors thank Jarrod Hadfield for providing guidance on implementing the MCMCglmm library.

## Author contributions

W.M.: conceptualisation, methodology, software, validation, formal analysis, data curation, writing–original draft, writing–review & editing, visualisation. G.R.: conceptualisation, methodology, validation, investigation, data curation, writing–review & editing. E.P.: methodology, writing–review & editing. M.C.: investigation, writing–review & editing. N.K.: investigation, writing–review & editing. L.B.: resources, writing–review & editing. M.M.: resources, writing–review & editing. S.O.: resources, writing–review & editing. K.L.H.: investigation, writing–review & editing. A.R.: writing–review & editing, project administration. D.K.: writing–review & editing. M.B.A.: writing–review & editing, supervision. A.S.W.: writing–review & editing, supervision, funding acquisition. S.L.: conceptualisation, writing–review & editing, supervision. N.S.: conceptualisation, writing–review & editing, supervision, project administration, funding acquisition.

## Competing interests

The authors declare no competing interests.
