## [Peer Review file · Nature Communications]

***Escherichia coli* phylogeny drives co-amoxiclav resistance through variable expression of TEM-1 beta-lactamase**

Corresponding Author: Dr William Matlock

Version 0:

Reviewer comments:

Reviewer #1

(Remarks to the Author)

In the manuscript titled “*E. coli* phylogeny drives co-amoxiclav resistance through variable expression of blaTEM-1” the authors look into the question of what possibly drives differential antibiotic resistance to co-amoxiclav, a commonly used drug combination, among a large collection of isolates which share a known antibiotic resistance gene.

This type of question is both important and challenging.

To address it, the authors use a combination of phylogenetic analysis and models based on gene content and gene expression measurements. Their results suggest that phylogeny may play a role.

In my opinion, the major thing missing is a transition from a statistical signal to a more tangible effect. Namely, how big is the predicted effect of phylogeny. Clarifying this point will improve this paper immensely.

Major points:

1. Figure 1 can contain more information. For example, I would suggest that data points in panel D be horizontally sorted the same as panel C. This would allow the reader to visually discern possible correlations between phylogeny and MIC. Currently the reader is left guessing. Also, it might be more informational to include details about the plasmids and their “phylogeny”. Such as dividing by origin etc.
2. For the long paragraph titled: *E. coli* phylogeny shapes blaTEM-1 expression starting in line 309.
 - 2.1 If I understand correctly, most of the contribution to expression comes from promoter mutations and copy number of the gene (line 343). Therefore, I do not fully understand what supports the title of the paragraph. Please clarify in the text.
 - 2.2 Also, it is later mentioned that phylogenies were “associated” with increased or decreased expression (lines 350-353). Are these associations done in separate from additional variables? Specifically, are they corrected for promoter sequence or copy number or anything else?
 - 2.3 Only a limited selection of promoter mutations are tested (line 314). Please explain why. Is this used to minimize the number of parameters/features? If so, then how were these mutations chosen? Lastly, it is important to know how the chosen mutations are correlated with phylogeny.
3. In Figure 2. You mention ST12 and ST372. I would suggest finding a better and clearer way to point to them. The outline is not enough. Also, is there a “follow up” for these marked groups? Reference to it in the text of the results or the discussion? I would definitely suggest some discussion of this result.
4. In the paragraph titled: Efflux pump AcrF is not encoded by all *E. coli* (starting at line 383). It seems that AcrF was chosen as a parameter for modelling later on. Choosing a specific gene as a parameter or model feature may result in bias or even sometimes be misleading. It is not clear why this gene was chosen. Please explain in the text.

Minor notes (mainly about jargon):

5. For me, “co-amoxiclav” is somewhat of a medical jargon. I believe the authors recognise it being jargon when they explicitly mention what co-amoxiclav is (line 56). I would suggest that the fact that this is a commonly used drug combination be mentioned in the abstract too.
6. A single sentence introduction to pOXA-48 (line 73) would also be helpful and allow smoother and clearer reading.

(Remarks on code availability)

Reviewer #2

(Remarks to the Author)

Evolution, function heterogeneity and adaptability of resistance genes in different bacterial clones are important scientific questions in the field of AMR research. This work utilized the widespread of beta-lactamase gene blaTEM-1 as a model to investigate its phenotypic heterogeneity in different E.coli clones. Authors found that E.coli phylogeny is the key factor influencing the MICs of co-amoxiclav. The findings are interesting and shed light on other resistance mechanisms research, especially in the field of resistance heterogeneity. However, the molecular mechanisms of controlling diverse expression of blaTEM-1 in various clones were not probed in details. Copy numbers and promoter difference may account for such phenotype to some extent, but it seems not enough to explain with full confidence. Although authors used various diverse genomics techniques, the following molecular mechanism investigation lacks.

(Remarks on code availability)

Reviewer #3

(Remarks to the Author)

Matlock and colleagues present a statistical analysis of genetic drivers of resistance to co-amoxiclav in E. coli. Through multivariate regression analysis they show how variation in TEM-1 gene copy number (primarily due to plasmid copy number variation) and TEM-1 expression (primarily due to promoter variants) largely determine MIC to co-amoxiclav. Moreover, the authors present data supporting the role of phylogenetic background as a mediator of resistance by influencing TEM-1 expression.

Strengths of this study include careful statistical treatment of known genetic features associated with resistance to understand causal links, and attempts to infer the independent role of genetic background on resistance. However, there are significant weaknesses that detract from enthusiasm, including the lack of discovery of novel mediators of resistance, the lack of insight into how genetic background might mediate TEM-1 expression, lack of support for variation in MIC above clinical breakpoints having clinical significance, some issues with data presentation and several speculative statements that aren't strongly supported by the data or citations.

Major comments

1. While the careful statistical modeling to tease apart the contributions of genotypes and phylogeny to MIC is excellent, it would help the reader if some of the raw data is also visualized to help provide more of an intuition for the signal the models are capturing. For instance, unless I missed it, I didn't see a figure showing MICs and TEM expression on the phylogeny. To tie everything together it would be helpful to have these two experimental measures, along with plasmid copy number and informative promoter variants overlaid on the phylogeny to enable the reader to evaluate the model in light of the data more clearly.

2. It isn't clear to me why the "phylogenetic effect" on MIC/expression in Figure 2 doesn't strictly align with phylogenetic structure (i.e. adjacent tips can vary widely in the contribution of phylogeny). Can the authors provide insight into this? Do these tips that defy phylogenetic expectation provide an opportunity to identify potential genetic mediators of phylogenetic signal? In addition, I think including MICs alongside the tree in Figure 2 would be helpful to the reader.

3. Was the plasmid carrying TEM evaluated as a mediator of phylogenetic signal?

4. There are several statements made in the discussion that aren't strongly supported by the data presented and/or need additional explanation/citation. For example, it is implied that increased expression of TEM-1 in clinically successful lineages may be causal for their success. Is there evidence that the MIC, once above the clinical breakpoint is positively associated with epidemic success in other contexts? In contrast to this hypothesis, it has been shown that increased expression of resistance genes is associated with fitness costs in vitro (note the authors themselves make this point on line 479, although don't provide the citations). A second example, the authors state this finding should generalize to other resistance genes, without providing any more nuanced discussion or support from the literature.

Minor comments

Line 431 – These => This

(Remarks on code availability)

Version 1:

Reviewer comments:

Reviewer #1

(Remarks to the Author)

Overall, the authors have addressed my comments to my satisfaction.

However, I strongly recommend that some version of their response to comment 2.3 (regarding the rationale for including only a limited set of promoter mutations in their model) be also incorporated into the manuscript. This would give readers a clearer understanding of the results and allow them to better individually assess the robustness of these findings.

(Remarks on code availability)

Reviewer #2

(Remarks to the Author)

No further comments.

(Remarks on code availability)

Reviewer #3

(Remarks to the Author)

I thank the authors for their responses and revisions made to the manuscript. However, I have some lingering concerns with lack of clear presentation of raw data, questions on interpretation of models as raised by other reviewers and questions regarding the new pan-genome analysis.

Major comments

1. Two/three reviewers asked for MICs to be included in main figures, to allow readers to understand the data more clearly when interpreting the results of the modeling. I think the authors response that the impact of phylogeny is subtle only accentuates why it is important for the reader to take this into account when interpreting the analysis. New Figure S5 showing all the requested data is quite challenging to interpret. Instead of (or in addition) Figure S5, I would appreciate showing MIC and TEM expression on the subset of genomes in Figure 2, alongside the model estimates of phylogenetic effect. This is the data that underlies the key findings from the paper, and would be nice for the reader to be able to see clearly.

2. I appreciate the author's identifying rep types for all circularized plasmids in the main text. Since the plasmid context of blaTEM is most critical to the results of this manuscript, can the putative rep types for the 277 circularized TEM plasmids, and distributions across phylogroups/STs, be provided? This would help the reader assess the potential role of plasmid/lineage associations in contributing to phylogenetic signal.

3. It is difficult from the results to understand the relative contribution of phylogeny to expression. From lines 364-375 and Table 2, it seems that the contribution of phylogeny is rather small compared to other variables. However, no significance is provided to support the contribution of phylogeny to expression, which is implied by the section title (E. coli phylogeny shapes blaTEM-1 expression).

4. How do the authors explain the disconnect between the first model, showing a modest association between phylogeny and expression, and the last model, showing that phylogenetic impact on MIC (a much stronger effect) is mediated by gene expression?

5. I have some concerns regarding the new pan-genome analysis. First, the core genome is much smaller than prior studies of E. coli that used much larger genome collections (e.g. 2,172 reported by Cummins et al., Microbial Genomics, 2022). How do the authors account for this discrepancy? The results from Piggy are quite surprising in terms of the tiny number of core IGRs. Is it possible that the sequence identity threshold is simply too strict to capture promoter evolution over large phylogenetic distances? Regardless, it is a pretty tenuous link with the rest of the manuscript, with the implication that variation in IGRs at this high level means that undetected variation in regulatory networks underlies the putative link between phylogeny and co-amoxiclav resistance.

(Remarks on code availability)

Version 2:

Reviewer comments:

Reviewer #3

(Remarks to the Author)

My prior critiques have now been addressed.

(Remarks on code availability)

Response to Reviewers

Reviewer #1 (Remarks to the Author):

In the manuscript titled “E. coli phylogeny drives co-amoxiclav resistance through variable expression of blaTEM-1” the authors look into the question of what possibly drives differential antibiotic resistance to co-amixoclav, a commonly used drug combination, among a large collection of isolates which share a known antibiotic resistance gene. This type of question is both important and challenging. To address it, the authors use a combination of phylogenetic analysis and models based on gene content and gene expression measurements. Their results suggest that phylogeny may play a role. In my opinion, the major thing missing is a transition from a statistical signal to a more tangible effect. Namely, how big is the predicted effect of phylogeny. Clarifying this point will improve this paper immensely.

Thank you for your review. We fully agree that this statistic would add value to our study, and have quantified it from line 466 as follows:

“We found that phylogenetic tip effects alone were sufficient to shift category membership across the EUCAST resistance breakpoint ($>8/2$ $\mu\text{g}/\text{mL}$). Fixing all other covariates at the midpoint of their latent categories, 19% (55/292) of isolates had a greater than 50% posterior probability of being shifted from the susceptible to the resistant category due to their phylogenetic effect alone. Of these, most were represented by phylogroup B2 and D at 64% (35/55) and 20% (11/55), respectively. The most represented sequence types were ST12, ST127, and ST69 at 27% (15/55), 20% (11/55), and 18% (10/55), respectively. The isolates in these sequence types also used in the expression model are highlighted in Figure 2a.

We have applied the EUCAST clinical breakpoint that is applicable to bloodstream infections (as our isolates were taken from bloodstream infection cases).

Major points:

1. Figure 1 can contain more information. For example, I would suggest that data points in panel D be horizontally sorted the same as panel C. This would allow the reader to visually discern possible correlations between phylogeny and MIC. Currently the reader is left guessing. Also, it might be more informational to include details about the plasmids and their “phylogeny”. Such as dividing by origin etc.

Thank you. We agree that more clarity is needed here, so have included an additional supplementary figure (Figure S4), which visualises the joint distribution of phylogroup and MIC. This figure is referenced on line 311 as follows:

“Figure S4 visualises the joint distribution of isolate phylogroup and MIC.”

However, it is crucial to note that any correlations in the raw data between phylogroup and MIC are highly confounded by the other covariates included in the models.

We have also added additional details regarding the *in silico* replicon typing for the plasmids in line 322 as follows:

“...*in silico* replicon typing revealed the most common plasmid families to be ColRNAI-like, Col156-like, B/O/K/Z-like and Col(MG828)-like at 11% (117/1,036), 7.4% (77/1,036), 6.8% (70/1,036), and 5.2% (54/1,036), respectively. All other plasmid families had fewer than 50 representatives.”

2. For the long paragraph titled: *E. coli* phylogeny shapes blaTEM-1 expression starting in line 309.

2.1 If I understand correctly, most of the contribution to expression comes from promoter mutations and copy number of the gene (line 343). Therefore, I do not fully understand what supports the title of the paragraph. Please clarify in the text.

Thank you for raising this point. While other covariates do explain a substantial fraction of the variance, our model additionally reveals a non-negligible contribution from the phylogenetic relatedness of the isolates, unexplained by the other covariates.

We used the term “shapes” deliberately to reflect that *E. coli* phylogeny contributes to, but does not fully determine, expression levels. Accordingly, we have added the following clarifying sentence to line 385:

“After accounting for these covariates, we still identified a contribution to variance from isolate phylogeny.”

2.2 Also, it is later mentioned that phylogenies were “associated” with increased or decreased expression (lines 350-353). Are these associations done in separate from additional variables? Specifically, are they corrected for promoter sequence or copy number or anything else?

This statement is based on the output from the model described in this section, specifically the individual tip effects, which are visualised in Figure 2b. We utilised a mixed model, where the effect of phylogeny on blaTEM-1 expression was estimated alongside the following covariates: qPCR replicate (as a random effect), blaTEM-1 cell and genome copy number, and presence and interaction of C32T and G175A promoter mutations (all fixed effects).

Phylogeny tip effects can be used as hypothesis generating tools, for example, to predict which isolates have the greatest phylogenetic effect on expression, but should not be directly tested upon due to their inherent uncertainty (e.g. PMID: 29622923). This

motivated the later use of our causal model (described in the section “*E. coli* phylogeny drives co-amoxiclav resistance through expression”), where we find that the effect of phylogeny on expression causally varies MIC.

2.3 Only a limited selection of promoter mutations are tested (line 314). Please explain why. Is this used to minimize the number of parameters/features? If so, then how were these mutations chosen? Lastly, it is important to know how the chosen mutations are correlated with phylogeny.

You are correct, this was to minimise the number of features in the model. We prioritised gathering expression data for isolates with a well-studied but rare mutation (C32T) and poorly-studied but common mutation in our dataset (G175A). The additional advantage of focussing on just two mutations was our ability to also explore the effect of their interaction (see Table 2).

We have included an additional supplementary figure (Figure S3) which visualises the distribution of phylogroup and promoter mutations. This figure is referenced on line 306 as follows:

“Figure S3 visualises the joint distribution of isolate phylogroup and linked *bla*_{TEM-1} promoter SNV.”

Again, it is important to note that any correlations in the raw data highly confounded by the other covariates.

3. In Figure 2. You mention ST12 and ST372. I would suggest finding a better and clearer way to point to them. The outline is not enough. Also, is there a “follow up” for these marked groups? Reference to it in the text of the results or the discussion? I would definitely suggest some discussion of this result.

Thank you for raising this. Based on our additional analysis above, we have updated these to instead include ST12, ST127, and ST69 (i.e., the isolates associated with crossing the EUCAST threshold).

We also agree that additional emphasis is needed in Figure 2, so have added additional coloured circles to the crown nodes for ST12, ST127, and ST69, and also chosen brighter colours. We have updated the Figure 2 legend accordingly.

Alongside mentioning these isolates in the results line 466 (as described above), we have updated the abstract the mention them in line 41:

“Phylogenetic effects alone were sufficient to shift MIC past the EUCAST breakpoint in 19% (55/292) of modelled isolates with a greater-than-chance probability. Most of these isolates represented ST12, ST69, and ST127.”

And referenced them in the Discussion in line 612:

“We also quantified that the clinically successful sequence types ST12, ST69, and ST127 had the highest probability of “flipping” an isolate from the sensitive to resistant category.”

4. In the paragraph titled: Efflux pump AcrF is not encoded by all *E. coli* (starting at line 383). It seems that AcrF was chosen as a parameter for modelling later on. Choosing a specific gene as a parameter or model feature may result in bias or even sometimes be misleading. It is not clear why this gene was chosen. Please explain in the text.

Thank you for raising this issue. Upon further investigation, we found an issue with the NCBI Reference Gene Catalog (<https://www.ncbi.nlm.nih.gov/pathogens/refgene/>), where only the AcrF sequence of the AcrEF–TolC efflux systems is present (i.e., missing the AcrE and TolC sequences). Upon further inspection, we found AcrEF–TolC systems in all $n=377$ isolates, though sometimes with less than 95% coverage and/or identity. In light of this, we have removed the “AcrF is not encoded by all *E. coli*” section from the Results and no longer use AcrF presence/absence as a model covariate. We have added the following to line 636:

“We also identified no relevant efflux pumps beyond AcrEF-TolC (found in every isolate), and it was out of the scope of the study to explore their functionality.”

We emphasise that removing AcrF from the models did not change the interpretation of any of the results, nor the conclusions of the study. This is because in the previous model, the AcrF covariate had no measurable effect (coefficient=0.02, 95% HPD=-0.51 – 0.54).

Minor notes (mainly about jargon):

5. For me, “co-amoxiclav” is somewhat of a medical jargon. I believe the authors recognise it being jargon when they explicitly mention what co-amoxiclav is (line 56). I would suggest that the fact that this is a commonly used drug combination be mentioned in the abstract too.

We have now specified upfront in the abstract that co-amoxiclav is a combination of amoxicillin-clavulanate and is commonly used clinically. We have added the following to the abstract in line 23:

“Co-amoxiclav (amoxicillin-clavulanate) is a commonly used combination antibiotic, with resistance in *E. coli* a clinically important phenotype associated with increased mortality.”

6. A single sentence introduction to pOXA-48 (line 73) would also be helpful and allow smoother and clearer reading.

We have updated this sentence in line 83 to define pOXA-48:

“Likewise, the introduction of a pOXA-48, a common conjugative plasmid in carbapenem-resistant clinical Enterobacterales, to six different *E. coli* strains resulted in variable co-amoxiclav resistance”

Reviewer #2 (Remarks to the Author):

Evolution, function heterogeneity and adaptability of resistance genes in different bacterial clones are important scientific questions in the field of AMR research. This work utilized the widespread of beta-lactamase gene blaTEM-1 as a model to investigate its phenotypic heterogeneity in different *E. coli* clones. Authors found that *E. coli* phylogeny is the key factor influencing the MICs of co-amoxiclav. The findings are interesting and shed light on other resistance mechanisms research, especially in the field of resistance heterogeneity. However, the molecular mechanisms of controlling diverse expression of blaTEM-1 in various clones were not probed in details. Copy numbers and promoter difference may account for such phenotype to some extent, but it seems not enough to explain with full confidence. Although authors used various diverse genomics techniques, the following molecular mechanism investigation lacks.

Thank you for your review. Our aim was not to identify specific novel molecular resistance mechanisms, but to quantify how known genetic factors operate in a real world clinical population. Controlled lab strain experiments often lack the genetic and ecological diversity of clinical isolates; by contrast, our multivariate hierarchical model integrates plasmid copy number and promoter variants across hundreds of bloodstream isolates, revealing how these drivers contribute to co-amoxiclav MIC. A major novelty of our study is being able to characterise the local (i.e. promoter associations, plasmid versus chromosomal location) as well as wider (i.e. lineage association) context of blaTEM-1 by being able to analyse nearly 400 fully reconstructed *E. coli* genomes. In doing this we have demonstrated that it is not only the combination of specific molecular mechanisms present that play a role on phenotypic resistance, but their joint interaction with the broader phylogenetic background.

We fully agree that uncovering the precise genetic elements responsible is a critical next step. Indeed, high-throughput mutational scanning approaches (e.g., QMS-seq in Jago et al., Nat Commun 2025) have only recently begun to quantify background effects on resistance evolution, with detailed experimental work on four strains of *E. coli* exposed to three different antibiotics identifying that 37% of resistance-associated mutations were found in intergenic regions, although these regions made up only 13% of the *E. coli* genome. Although implementing such screens across diverse clinical lineages is beyond this study's scope, to bridge this gap we have supplemented our original analyses with a comprehensive survey of intergenic region (IGR) architecture (from line 551). In brief, we performed parallel pangenome analyses of 15,781 gene clusters and 33,345 IGR

clusters across our 377 chromosomes, revealing that IGR diversity outpaced coding-sequence diversity in a phylogroup-specific manner (Figure 3). Specifically, cluster accumulation curves and Heaps' law fits showed that IGRs accumulated novelty more rapidly than genes, most prominently in phylogroup A.

Our substantial additional analysis reveals that *E. coli* phylogroups differ in intergenic architecture and dynamics of genetic flow. These lineage-specific differences in the non-coding "regulome" potentially underlie the phylogenetic signal we observe in *bla*_{TEM-1} expression. Future studies should therefore not only consider coding regions, but also target intergenic regions to fully capture other regulatory mechanisms that may be influencing AMR gene expression, for example trans-acting transcription factors, small regulatory RNAs, DNA methylation patterns, and nucleoid-associated proteins. We have updated the discussion accordingly from line 666:

"We posit that lineage-specific regulatory systems in *E. coli*, shaped by horizontal gene flow, may house the key modulators of *bla*_{TEM-1} expression. Although comprehensive dissection of these elements (for example, through ChIP-seq) extends beyond the scope of this study, our findings suggest that future efforts directed at mapping trans-acting factors, small RNAs, differential DNA methylation, and nucleoid-associated protein binding across phylogroups will be essential."

Reviewer #3 (Remarks to the Author):

Matlock and colleagues present a statistical analysis of genetic drivers of resistance to co-amoxiclav in *E. coli*. Through multivariate regression analysis they show how variation in TEM-1 gene copy number (primarily due to plasmid copy number variation) and TEM-1 expression (primarily due to promoter variants) largely determine MIC to co-amoxiclav. Moreover, the authors present data supporting the role of phylogenetic background as a mediator of resistance by influencing TEM-1 expression.

Strengths of this study include careful statistical treatment of known genetic features associated with resistance to understand causal links, and attempts to infer the independent role of genetic background on resistance. However, there are significant weaknesses that detract from enthusiasm, including the lack of discovery of novel mediators of resistance, the lack of insight into how genetic background might mediate TEM-1 expression, lack of support for variation in MIC above clinical breakpoints having clinical significance, some issues with data presentation and several speculative statements that aren't strongly supported by the data or citations.

Thank you for your review. As per our response to Reviewer 2 above, the strength and major novelty of our study lies in quantifying the impact of multiple mechanisms on co-amoxiclav resistance in a large dataset of clinically relevant isolates, complementing

studies using controlled experiments in strains which seek to characterise in detail the impact of specific mechanisms in a fixed but typically unrepresentative genetic background.

In addition to this novelty in quantifying the impact of known molecular mechanisms of co-amoxiclav resistance on expression and MIC, we have also determined that the association of these known mechanisms is influenced by the phylogenetic context, which has not, to our knowledge, been considered in such a large dataset representative of the population diversity of infection-associated *E. coli* previously. To address the Reviewer's concern that our analysis showed a "lack of insight into how genetic background might mediate TEM-1 expression," we have now expanded our analysis by considering variability in intergenic regions (IGRs; from line 551), shown to be highly relevant to AMR (in PMID: 39824824), with the relevance of this described above.

We disagree with the criticism of "lack of support for variation in MIC above clinical breakpoints having clinical significance". As outlined in the paragraph starting line 64, isolates above the EUCAST clinical breakpoint (8/2 µg/mL) seem to contribute to important differences in clinical outcome, as co-amoxiclav treatment of *E. coli* bacteraemia with MICs > 32/2 µg/mL has been shown to be associated with significantly higher mortality in *E. coli* bacteraemia (PMID: 35723965)". Therefore, we believe considering MICs above 8/2 µg/mL is relevant, and a major strength of this study.

We have addressed the issues with data presentation below, as well as clarifying any statements throughout.

Major comments

1. While the careful statistical modeling to tease apart the contributions of genotypes and phylogeny to MIC is excellent, it would help the reader if some of the raw data is also visualized to help provide more of an intuition for the signal the models are capturing. For instance, unless I missed it, I didn't see a figure showing MICs and TEM expression on the phylogeny. To tie everything together it would be helpful to have these two experimental measures, along with plasmid copy number and informative promoter variants overlaid on the phylogeny to enable the reader to evaluate the model in light of the data more clearly.

We agree that further visualisation of the raw data is a valuable addition. However, we also caution that any correlations in the raw data are greatly confounded by the other covariates.

Alongside the new supplementary Figures S3 and S4, we have also visualised the distribution of cell and genome *bla*_{TEM-1} copy number, co-amoxiclav MIC, and *bla*_{TEM-1} expression against the phylogeny in Figure S5, referencing it in line 312:

“Figure S5 visualises the distribution of *bla*_{TEM-1} genome and cell copy number, *bla*_{TEM-1} expression, and co-amoxiclav MIC against the core-gene chromosomal phylogeny.”

2. It isn't clear to me why the “phylogenetic effect” on MIC/expression in Figure 2 doesn't strictly align with phylogenetic structure (i.e. adjacent tips can vary widely in the contribution of phylogeny). Can the authors provide insight into this? Do these tips that defy phylogenetic expectation provide an opportunity to identify potential genetic mediators of phylogenetic signal? In addition, I think including MICs alongside the tree in Figure 2 would be helpful to the reader.

This is a limitation of our methodology, and accordingly, have added the following to the Discussion on line 638:

“Fifthly, we found that tip effects did not strictly align with phylogenetic structure. We used a single fixed phylogeny and a single global phylogenetic variance parameter, which does not capture uncertainty in tree topology or clade-specific evolutionary rates. Consequently, tips within densely sampled clades might be over-shrunk toward their mean, while tips in sparse clades might be under-shrunk. Future work should incorporate phylogenetic uncertainty (e.g. sampling from a tree posterior) and explore multi-variance or partitioned phylogenetic models to assess the impact of tree balance on tip-effect estimation”

As described above, MICs are now plotted against phylogeny in Figure S5.

3. Was the plasmid carrying TEM evaluated as a mediator of phylogenetic signal?

This is a really interesting consideration, and highlights a limitation of our approach. The problem is that the plasmid population carrying *bla*_{TEM-1} in clinically isolates is highly diverse: In total, we identified $n=138$ unique replicon/multi-replicons, of which 62% (85/138) had a least one representative carrying *bla*_{TEM-1} (from Table S1). Incorporating phylogenies for all of these plasmid groups in the model would be computationally prohibitive and statistically unstable with currently available methods that we are aware of. Whilst future studies could focus on curating datasets with less plasmid diversity, this would then not be reflective of true clinical bacterial populations. Accordingly, we have added the following to the Discussion in line 645:

“Sixthly, modelling phylogenetic effect terms for all the plasmid replicons in the dataset would be computationally prohibitive and statistically unstable. Whilst future studies could focus on curating datasets with less plasmid diversity, this would not be reflective of true clinical bacterial populations.”

4. There are several statements made in the discussion that aren't strongly supported by the data presented and/or need additional explanation/citation. For example, it is implied that increased expression of TEM-1 in clinically successful lineages may be causal for

their success. Is there evidence that the MIC, once above the clinical breakpoint is positively associated with epidemic success in other contexts? In contrast to this hypothesis, it has been shown that increased expression of resistance genes is associated with fitness costs in vitro (note the authors themselves make this point on line 479, although don't provide the citations). A second example, the authors state this finding should generalize to other resistance genes, without providing any more nuanced discussion or support from the literature.

We agree that speculative statements should always be clearly delineated from results. However, our Discussion simply states that the traits of higher expression and MIC were associated with some clinically epidemic strains. We have added additional text to clarify that any further interpretation is purely speculative in line 673:

“Our study demonstrates the some clinically successful lineages are better at expressing *bla*_{TEM-1}, and have a higher probability of “flipping” from sensitive to resistant. We speculate that, since *bla*_{TEM-1} is both widespread in clinical bacterial populations and its spread is plasmid mediated, being able to better control the expression of *bla*_{TEM-1} when acquired is a selective advantage for clinical isolates.”

We have removed the statement that our findings will generalise to other AMR genes, but we believe our approach represents a relevant framework for considering other AMR genotype-phenotype combinations

Minor comments

Line 431 – These => This

Thank you for noticing this, we have updated it in line 603.

Reviewer #1 (Remarks to the Author):

Overall, the authors have addressed my comments to my satisfaction. However, I strongly recommend that some version of their response to comment 2.3 (regarding the rationale for including only a limited set of promoter mutations in their model) be also incorporated into the manuscript. This would give readers a clearer understanding of the results and allow them to better individually assess the robustness of these findings.

We agree this is useful context, so have added it in line 343:

“Moreover, we only selected isolates with zero, one, or two mutations in the *bla*_{TEM-1} promoter sequence: C32T, a well-studied but rare mutation which produces two overlapping promoters and is known to increase expression, and G175A, a less studied but common mutation in our dataset (according to Sutcliffe numbering based on the PBR322 plasmid; see Table 1). Focussing on only two mutations enabled us to statistically explore the effect of their interaction.”

Reviewer #2 (Remarks to the Author):

No further comments.

Reviewer #3 (Remarks to the Author):

I thank the authors for their responses and revisions made to the manuscript. However, I have some lingering concerns with lack of clear presentation of raw data, questions on interpretation of models as raised by other reviewers and questions regarding the new pan-genome analysis.

Major comments

1. Two/three reviewers asked for MICs to be included in main figures, to allow readers to understand the data more clearly when interpreting the results of the modeling. I think the authors response that the impact of phylogeny is subtle only accentuates why it important for the reader to take this into account when interpreting the analysis. New Figure S5 showing all the requested data is quite challenging to interpret. Instead of (or in addition) Figure S5, I would appreciate showing MIC and TEM expression on the subset of genomes in Figure 2, alongside the model estimates of phylogenetic effect. This is the data that underlies the key findings from the paper, and would be nice for the reader to be able to see clearly.

We agree and have incorporated raw MIC and expression data as an additional panel in Figure 2. We have updated the legend accordingly:

“(b) Bar length records mean *bla*_{TEM-1} Δ Ct for each tip (log₁₀-scaled; lower values denote

higher *bla*_{TEM-1} expression). Bar colour records isolate co-amoxiclav MIC for each tip.”

We believe that this addition, alongside the existing Figure 1, gives the reader a well-rounded view of the raw data.

2. I appreciate the author’s identifying rep types for all circularized plasmids in the main text. Since the plasmid context of *bla*_{TEM} is most critical to the results of this manuscript, can the putative rep types for the 277 circularized TEM plasmids, and distributions across phylogroups/STs, be provided? This would help the reader assess the potential role of plasmid/lineage associations in contributing to phylogenetic signal.

We agree that this would be a valuable addition. Alongside the raw data for these distributions in Supplementary Table 1, they are now plotted in Supplementary Figure 6. This is detailed in line 315:

“Figure S6 visualises the relationship between strain background (sequence type and phylogroup) and replicon types (PlasmidFinder output; see Materials and methods) for the circularised plasmids.”

We have also updated the Supplementary Figure legends document accordingly.

3. It is difficult from the results to understand the relative contribution of phylogeny to expression. From lines 364-375 and Table 2, it seems that the contribution of phylogeny is rather small compared to other variables. However, no significance is provided to support the contribution of phylogeny to expression, which is implied by the section title (*E. coli* phylogeny shapes *bla*_{TEM-1} expression).

We appreciate the opportunity to clarify. We agree that the mean phylogenetic variance (~7%) is modest, but its broad posterior distribution (ranging up to ~21%) reflects substantial heterogeneity across the tree. This right-skewed distribution indicates that while most strains contribute a modest amount, deeper splits in the phylogeny capture meaningful expression divergence among certain lineages. In Bayesian models, such tail weight suggests non-negligible structure even when the average effect is small. We therefore maintain that phylogeny exerts a measurable, if uneven, influence on *bla*_{TEM-1} expression.

We have updated the text from in line 381 to reflect this:

“The posterior for contribution of variance from phylogeny demonstrated a long right tail (mean=0.07; 95% highest posterior density, HPD=[0.00, 0.21]; see Materials and methods), suggestive of heterogeneity in phylogenetic signal, where deeper splits between major lineages may explain disproportionately large differences in expression”

4. How do the authors explain the disconnect between the first model, showing a modest

association between phylogeny and expression, and the last model, showing that phylogenetic impact on MIC (a much stronger effect) is mediated by gene expression?

The difference in contributions to variance from expression and MIC reflect the fact that the two traits live on very different scales, so the raw variance components from both models are not directly comparable. This was the motivation for combining both traits into a third, unified model.

5. I have some concerns regarding the new pan-genome analysis. First, the core genome is much smaller than prior studies of *E. coli* that used much larger genome collections (e.g. 2,172 reported by Cummins et al., *Microbial Genomics*, 2022). How do the authors account for this discrepancy? The results from Piggy are quite surprising in terms of the tiny number of core IGRs. Is it possible that the sequence identity threshold is simply too strict to capture promoter evolution over large phylogenetic distances? Regardless, it is a pretty tenuous link with the rest of the manuscript, with the implication that variation in IGRs at this high level means that undetected variation in regulatory networks underlies the putative link between phylogeny and co-amoxiclav resistance.

We thank the reviewer for raising these points. Our analysis differs from Cummins et al. (2022) in that they constructed pangenomes within specific *E. coli* sequence types (STs), each of which is far more genetically homogeneous than our cross-phylogroup dataset. When one collapses all phylogroups into a single pangenome, the proportion of truly core genes necessarily shrinks. The ~5% fraction we observed (783 gene clusters found in every isolate out of 15,781 total clusters) is in line with other broad *E. coli* pangenome studies spanning multiple phylogroups (e.g. PMID: 29077859 determined ~4% with 500 cross-phylogroup *E. coli* isolates).

Piggy's default 90% nucleotide-identity and length-identity cut-off is indeed conservative. We identified 322 core IGRs (i.e. those found in 377/377 isolates), roughly one per adjacent core gene pair, which is reasonable considering the pangenome is constructed cross-phylogroup, and *E. coli* is subject to frequent structural rearrangements (e.g. <https://doi.org/10.1101/2024.07.08.602537>). Reducing the threshold (e.g. to 80%) risks grouping non-orthologous IGRs and inflating core counts, which would obscure genuine lineage-specific differences in promoter sequences.

Bacterial promoters reside in intergenic regions, and our pangenome survey shows that IGRs accumulate diversity far more rapidly than coding regions (particularly in phylogroup A) thereby creating lineage-specific promoter landscapes. Recent QMS-seq data (Jago et al., *Nat Commun*, 2025) demonstrate that 37% of resistance-associated mutations map to IGRs, despite these regions comprising only ~13% of the genome. This combination of rapid regulatory turnover and proven functional importance underscores the central role of noncoding variation in linking phylogenetic history to antibiotic resistance, which we believe should be the focus of future studies.